# XRN2 Is Required for Cell Motility and Invasion in Glioblastomas

**DOI:** 10.3390/cells11091481

**Published:** 2022-04-28

**Authors:** Tuyen T. Dang, Megan Lerner, Debra Saunders, Nataliya Smith, Rafal Gulej, Michelle Zalles, Rheal A. Towner, Julio C. Morales

**Affiliations:** 1Department of Neurosurgery, Sttephenson Cancer Center University of Oklahoma Health Science Center, Oklahoma City, OK 73104, USA; tuyen-dang@ouhsc.edu; 2Department of Surgery, University of Oklahoma Health Science Center, Oklahoma City, OK 73104, USA; megan-lerner@ouhsc.edu; 3Department of Pathology, University of Oklahoma Health Science Center, Advanced Magnetic Resonance Center, Oklahoma Medical Research Foundation, Oklahoma City, OK 73104, USA; debra-saunders@omrf.org (D.S.); nataliya-smith@ouhsc.edu (N.S.); rafal-gulej@ouhsc.edu (R.G.); mzalles@mdanderson.org (M.Z.); rheal-towner@omrf.org (R.A.T.)

**Keywords:** XRN2, cell motility, invasion, tumor progression

## Abstract

One of the major obstacles in treating brain cancers, particularly glioblastoma multiforme, is the occurrence of secondary tumor lesions that arise in areas of the brain and are inoperable while obtaining resistance to current therapeutic agents. Thus, gaining a better understanding of the cellular factors that regulate glioblastoma multiforme cellular movement is imperative. In our study, we demonstrate that the 5′-3′ exoribonuclease XRN2 is important to the invasive nature of glioblastoma. A loss of XRN2 decreases cellular speed, displacement, and movement through a matrix of established glioblastoma multiforme cell lines. Additionally, a loss of XRN2 abolishes tumor formation in orthotopic mouse xenograft implanted with G55 glioblastoma multiforme cells. One reason for these observations is that loss of XRN2 disrupts the expression profile of several cellular factors that are important for tumor invasion in glioblastoma multiforme cells. Importantly, XRN2 mRNA and protein levels are elevated in glioblastoma multiforme patient samples. Elevation in XRN2 mRNA also correlates with poor overall patient survival. These data demonstrate that XRN2 is an important cellular factor regulating one of the major obstacles in treating glioblastomas and is a potential molecular target that can greatly enhance patient survival.

## 1. Introduction

Glioblastoma multiforme tumors (GBMs) are lethal brain tumors, as they are highly aggressive in both growth rate and invasiveness. The five-year survival rate for GBM patients is <5% [1], with a median survival rate of 16 months [2]. GBMs are primarily treated with tumor resection, followed by radio- and chemo-therapies [3]. One of the major obstacles in treating GBMs is their highly invasive nature, exhibiting itself in extensive tumor infiltration into the surrounding parenchyma [4]. This invasive nature of GBMs results in tumor recurrence, with limited surgical treatment options, and acquisition of radio- and chemo-resistance in nearly 100% of patients [5]. 

The first step in GBM progression is the movement of a few tumor cells away from the primary tumor (dissemination), followed by the colonization and formation of a tumor in a secondary site within the brain [6]. GBM dissemination into the surrounding brain regions is a complex multi-step process [7] that depends on two characteristics: (1) the inherent movement of tumor cells (cell motility) and (2) the ability of tumor cells to migrate through a matrix (cell invasion). Thus, gaining a better understanding of the means driving cell motility and invasion is important for understanding the mechanisms behind GBM secondary site formation. Our current study demonstrates that XRN2 is a regulator of GBM aggressiveness. 

XRN2 is a 5′-3′ exoribonuclease that participates in several cellular processes. Classically, XRN2, the human homolog of the yeast protein RAT1, plays a role in transcription termination [8]. Moreover, XRN2 participates in transcription elongation by mediating premature termination [9]. In addition to having a function in transcription, XRN2 is also required for maintaining genomic stability, double-strand break repair, and RNA:DNA hybrid (R-loop) resolution [10,11]. Furthermore, XRN2 may affect additional cellular processes indirectly through an extended role in gene regulation by mediating R-loop resolution [12] and miRNA formation [13]. In fact, it has been shown that XRN2 can mediate the epithelial–mesenchymal transition (EMT) in lung cancer by regulating the maturation of miR-10a [13].

In this study, we demonstrate a role for XRN2 in mediating cell motility and migration through a matrix in GBM cells. We found that a loss of XRN2 can lead to decreased speed and displacement of U87 and U251 GBM cells. Along with this, we found that loss of XRN2 impaired the ability of GBM cells to migrate through a solid matrix. Strikingly, a loss of XRN2 resulted in a ~90% reduction in tumor volume when G55 cells were injected into mouse brains. Lastly, we demonstrate that XRN2 alters the transcriptional profile of GBM cells, suggesting that XRN2 can mediate the expression of cellular factors for cell motility and invasion in GBMs similar to the observations in lung cancer [13]. 

## 2. Materials and Methods

### 2.1. G55 Xenograft Model and Treatment

All animal studies were conducted with the approval of the Oklahoma Medical Research Foundation and University of Oklahoma Health Sciences Center Institutional Animal Care Use Committee policies, which follow NIH guidelines. Human G55 xenograft cells (modified and unmodified) were injected intracerebrally in two-month-old male mice (Hsd: Athymic Nude-Foxn1nu mice; Harlan Inc., Indianapolis, IN, USA), as previously described [14]. All mice were euthanized when tumors reached ~150 mm^3^, or up to 27 days post-implantation of G55 cells. 

### 2.2. Magnetic Resonance Imaging

Mice were anesthetized and positioned in a cradle that was inserted into a 30 cm horizontal bore Bruker Biospin magnet operating at 7 Tesla (Bruker BioSpin GmbH, Karlsruhe, Germany). All MRI experiments were carried out using the Bruker BA6 gradient set and mouse head coil, as previously described [14]. All animals were imaged every 3–4 days until the end of the study starting at 10 days post-G55 implantation surgery.

### 2.3. Patient Gene Expression

XRN2 mRNA expression in normal and glioblastoma patient samples and patient survivorship were generated using R2: Genomics Analysis and Visualization Platform using default program settings. The following databases were used: Berchtold [15], Pfister [16], Loeffler [17], Hegi [18], Kawaguchi [19], French [20], and Sun [21]. These data were accessed from 2018 to 2021. XRN2 mRNA expression across brain tumors was generated using Oncomine with the Sun database [21] accessed in September 2016.

### 2.4. Cell Culture 

Cells were maintained in a humidified 37 °C environment supplemented with 5% CO_2_. Growth media for each of the cell lines were as follows: 10% fetal bovine serum (FBS) in DMEM for G55 cells, 10% FBS in MEM supplemented with 2 mM L-glutamine and 1 mM sodium pyruvate for U251 and U87 cells, and 5% FBS in DMEM for LN229. All growth media were supplemented with 1X penicillin–streptomycin. 

### 2.5. H2B-GFP Labeling and shRNA Knockdown of Cells

Over-expression of histone 2B GFP fusion protein was achieved through transduction of cells with virus produced from transfection of pCLNR-H2BG (plasmid #17735, Addgene) in 293GP with respective virial packaging plasmids [22]. Shluciferase, shXRN2-3640, and shXRN2-3639 cells were generated through transduction with virus produced from transfection of the listed below plasmids from Sigma Aldrich into 293T with respective virial packaging plasmids [22].

### 2.6. VCL siRNA

VCL #1: GCAUUCAGGCCUCAGUGAA

VCL #2: GCAUAGAGGAAGCUUUAA

VCL #3: CAAGAUGAUUGACGAGAGA

### 2.7. Shluciferase (sku #SHC007): Proprietary Sequence

shXRN2-3640 (TRCN0000293640): CCGGGTGTATTCTAGATCATCTAAGCTCGAGCTTAGATGATCTAGAATACACTTTTTG

shXRN2-3639 (TRCN0000293639): CCGGTACATAGCTGATCGTTTAAATCTCGAGATTTAAACGATCAGCTATGTATTTTTG

### 2.8. Western Blot

Cells were lysed in RIPA buffer supplemented with a protease inhibitor cocktail. Samples were denatured in Laemmli buffer, ran on a 10% SDS–PAGE gel, and transferred on a PVDF membrane. Membranes were probed with XRN2 antibody (cat no. A301-103A, Bethyl, Montgomery, TX, USA) and GAPDH (cat no. ab181602, Abcam, Boston, MA, USA). Alexa-488 or Alexa-680 secondary antibodies were used to visualize expression on the Chemidoc MP (BioRad, Herules, CA, USA). 

### 2.9. siRNA Transfection/RT-PCR

Cells were reverse transfected with 50 nM of siRNAs (listed in ref) using MISSION siRNA transfection reagent (cat no. S1452-1ML, Sigma Aldrich, St. Louis, MO, USA). Then, 72 h after transfection, the total RNA was harvested using a miRNeasy Micro kit (cat no. 217084, Qiagen, Germantown, MD, USA) following manufacturer’s protocol. Real-time PCR (RT-PCR) was used to verify knockdown. Briefly, 500 ng of total RNA was converted into cDNA using iScript™ cDNA Synthesis Kit (cat no. 1708890, BioRad) following manufacturer’s protocol. BioRad CFX 96 Real-Time System, iTaq™ Universal SYBR^®^ Green Supermix (cat no. 1725125), and XRN2 and GAPDH primers (sequences listed below) were used for RT-PCR. Expression change was determined with GAPDH used as a loading control and fold change in ΔCt.

XRN2 forward primer: CAGCAACTGATTACACCAG

XRN2 reverse primer: ACTGTCAATTTTTCCACCC

GAPDH forward primer: CTTTTGCGTCGCCAG

GAPDH reverse primer: TTGATGGCAATATCCAC

### 2.10. RNA-Seq 

Total RNA was submitted to the core facility OUHSC Laboratory for Molecular Biology and Cytometry Research for library build, RNA sequencing, genome alignment, and quality analysis. RNAseq libraries were constructed using the Illumina TruSeq RNA LT v2 kit and established protocols. The library construction was carried out using total RNA isolated from human GBM cell lines (1 ug). RNA quality for each prep was analyzed prior to construction using the Agilent Bioanalyzer 2100 and nano total RNA chips. Each library was indexed during library construction in order to multiplex for sequencing on the Illumina MiSeq platform. Samples were normalized and sequenced in batches of 3 libraries per 2 × 150 bp for paired-end sequencing run on the Illumina MiSeq. On average, 44 million reads (7 Gb) of sequencing data were collected per run. Raw data for each sample were analyzed using CLC Genomics Workbench version 10.0.1 software from Qiagen (formerly CLCBio). Raw sequence reads were mapped to the Homo sapiens genome to identify genes expressed under each condition. A pairwise comparison of the expression results were performed using the total mapping results which specified experimental groups. Differential gene lists were created with 1.3-fold expression cutoff and significant *p* values of ≤0.05 to identify genes that were up- or down-regulated under each condition. 

### 2.11. Live Cell Imaging 

Cells were reverse transfected for 48 to 72 h. Cells were fed with fresh 1% FBS growth media supplemented with 100 ng/mL epidermal growth factor before imaging. Imaging was conducted on the LSM710 (Ziess) from the Oklahoma Research Medical Foundation Imaging core. The microscope was stored in a humidified, temperature- and CO_2_-controlled chamber. Images were taken with a 10× objective taken at 30 min intervals for at least 6 h. TrackMate [23] was used to track the movement of cells overtime.

### 2.12. Inverted Vertical Invasion Assay (Up-Invasion)

Cells were either reverse-transfected with siRNAs or transduced with shRNAs, and were plated in a 96-well plate at a density where the wells were ~100% confluent by 48 h. A mixture of 50:50 collagen I–Matrigel were plated at a volume of 50 μL per well on top of the monolayer. After the matrix had solidified (~1 h), 150 μL of growth media was overlayed. Cells were allowed to invade for 48 h before fixation with 2% formaldehyde diluted in 1X PBS for 20–40 min. Wells were then washed and stored in 1X PBS. Cultures were imaged on the LSM710 confocal using a 10× objective. Z-stacks were obtained by imaging 50 micron below and 100 micron above the monolayer with 5 micron slices. Invading cells were cells that had invaded 50+ micron above the monolayer. The extent of invasion was normalized to control conditions [24].

### 2.13. Immunohistochemistry/H&E

Human patient samples were taken from a commercial tissue microarray (cat no. BS17017b, US Biomax, Rockville, MD, USA). Patient samples were stained with 1:2000 dilution of XRN2 antibody (A301-103A, Bethyl Laboratories, Montgomery, TX, USA). Briefly, samples were baked at 60 °C for 30 min follow by xylene and alcohol washes. Subsequently, pH 6 citrate antigen retrieval was carried out for 20 min, peroxidase block was carried out at room temperature for 10 min, blocking was carried out for 20 min with 2.5% horse serum, and incubation with primary antibodies was carried out for 60 min at room temperature. The Impress Excel-Amplified Polymer Stain Anti-Rabbit Peroxidase kit (MP7601, Vector Labs, San Francisco, CA, USA) and Nova Red, as the chromogen, were used to visualize XRN2 staining, as per manufacturer’s instructions. Tumors from xenografts were fixed in formalin and then embedded in paraffin. Afterwards, tumors were sliced 5 microns thick for hematoxylin and eosin staining (H&E). Briefly, samples were washed in xylene and alcohol washes; incubated in Harris hematoxylin, acidic alcohol, and water washes for 15 min; and subjected to eosin staining, followed by alcohol and xylene washes. Images were captured on the Cytation 5 imaging system (BioTek, Winooski, VT, USA).

### 2.14. XRN2 Immunohistochemistry Staining Quantification

To quantify XRN2 expression, stained tissue sections were scanned on the Aperio Scanscope CS System (Aperio Technologies, Inc., Vista, CA, USA) and analyzed using the Aperio ImageScope program specifically the Positive Pixel Count 2004-08-11 algorithm. Positive XRN2 pixel staining was determined by a medium to strong pixel intensity. Positivity was determined by the number of positive pixels divided by total pixels and then multiplied by 100. A value of 2 positivity was considered as positive XRN2 staining. Normal brain tissues had positivity values between 0.31 and 1.37.

## 3. Results

### 3.1. XRN2 Display Elevated Expression in Gliomas

In recent years, XRN2 has emerged as a regulator of cancer cell migration in lung cancer and oral carcinoma [13,25]. To determine if XRN2 may also contribute to the progression of GBMs, we examined the Pfister, Loeffler, and Hegi glioma gene expression datasets for XRN2 expression, using the R2: Genomic Analysis and Visualization Platform (http://r2.amc.nl, accessed 1 March 2019). We found that XRN2 mRNA levels were elevated in glioma samples as compared to normal brain samples (Berchtold) (Figure 1A). This mRNA elevation of XRN2 suggests that XRN2 may play a role in GBM disease and may also be a potential biomarker for the disease.

To determine if XRN2 has a functional role in GBMs, we examined XRN2 protein expression in patient samples. Using a glioma tissue microarray (product number GL803c) purchased from US Biomax, we examined XRN2 protein expression levels through immunohistochemistry (IHC). The microarray contained 30 GBM, 28 astrocytoma, 9 oligodendroglioma, 4 oligoastrocytoma, and 4 normal brain tissue samples. Consistent with the elevation of XRN2 mRNA, we found increased XRN2 protein expression in GBM patient samples as compared to normal brain tissues (Figure 1B and Appendix A). Using the Aperio ImageScope (Leica Biosystems), we quantified a XRN2 IHC 2% or greater positivity staining in 22 of the 30 GBM samples, while no normal brain sample displayed 2% positivity staining (Figure 1C). Interestingly, we found that XRN2 mRNA and protein levels were elevated in GBMs, as compared to astrocytoma and oligodendrogliomas (Appendix A). Lastly, using the Kawaguchi data, we found that increased XRN2 mRNA levels correlated with poor overall patient survival, as compared to patients with lower XRN2 expression (Figure 1D). The poor survivorship suggests that XRN2 may not only play a role in GBM tumors, but also in the progression of the disease. 

### 3.2. XRN2 Mediates GBM Cell Motility

As XRN2 has been shown to mediate epithelial-derived tumor migration, we wanted to know if XRN2 may also drive the motility of non-epithelial cancers, such as GBMs. We examined how XRN2 loss affected the migration of glioma cells. To examine this, we employed U87 glioma cell lines modified to express histone 2B fused with green fluorescent protein (U87-GFP) (Figure 2A). We exposed U87-GFP cells to control and XRN2 siRNAs (Appendix A). Using live-cell imaging, we monitored single-cell movement in U87 cells with and without XRN2 (Figure 2A,B; Appendix A. We found that loss of XRN2 resulted in a ~25% decrease in speed and displacement in U87 cells (Figure 2C), as compared to control cells. Yet, this decrease was not the same as the ~50% decrease found in cells exposed to cytochalasin D (cytoD) (Figure 2C). We found a similar ~25% decrease in speed and displacement in U251 GBM cells (Appendix A). Based on these results, XRN2 is required for the intrinsic motility of GBM cells.

### 3.3. Loss of XRN2 Impairs GBM Invasiveness

Even though cell motility is a required step in metastasis, it is not sufficient to induce invasion. Thus, to determine if XRN2 is required for invasion, we examined if XRN2 mediates glioma cell invasion through a matrix. We utilized the inverted vertical invasion assay to measure invasiveness through a matrix [26]. To perform this inverted vertical invasion assay, we plated previously described control and XRN2-deficient G55 GBM cell lines [10] in a 96-well plate. After 48 h, we layered a collagen I–Matrigel matrix over the cells. After the matrix was solidified, we overlayed media and allowed the cells to grow and invade for another 48 h (Figure 3A). We then fixed the cells and used the LSM710 confocal microscope for image invasion of control and XRN2-deficient G55 cells (Figure 3B). We found that the loss of XRN2 resulted in a ~50% decrease in the relative invasion of two different XRN2-deficient cell lines (shXRN2-3640 and shXRN2-3639), as compared to control (shluc) cells (Figure 3C). Relative invasion was determined by dividing the number of cells that traveled ≥ 50 μm above the monolayer in XRN2-deficient cell lines by the number of cells that traveled > 50 μm above the monolayer in control samples [24]. The requirement of XRN2 in invasion was repeated when we used siRNAs to down-regulate XRN2 expression (Appendix A). 

### 3.4. Loss of XRN2 Diminishes GBM Growth In Vivo

To extend the observations made in vitro, G55 cells with or without XRN2 [10] were orthotopically injected into mouse brains for tumor xenograft studies. We used G55 GBM cells, as they are highly aggressive and readily form tumors in vivo when injected into mouse brains [14,27]. We injected six mice with control G55 cells, seven mice with G55-shXRN2, and four mice with G44-shXRN2 #2 cell. Using histological examination, we found that a loss of XRN2 reduced the tumor size visualized through the cross-sectioning of mouse brains (Appendix A). Consistent with histological data, using magnetic resonance imaging, we found that the tumor volume of two unique XRN2-deficient G55 cells was 5.85 and 2.85 mm^3^ (G55-shXRN2 (shXRN2-3640) and G55-shXRN2 #2 (shXRN2-3639), respectively), as compared to the 98.8 mm^3^ volume observed with control cells (Figure 4). We also found that a loss of XRN2 decreased the relative invasion in XRN2 LN229-deficient cells as compared to the control cells (Appendix A). Importantly, we have shown that a loss of XRN2 does not affect proliferation or elicit cell-cycle changes in G55 cells [10], consistent with other published reports [28,29]. Based on these results, XRN2 seems to play a vital role in tumor development. 

### 3.5. Loss of XRN2 Alters the Transcriptional Profile of Glioma Cells

Previous studies found that XRN2 plays a role in regulating miRNA maturation [13] and RNA:DNA hybrid formation [10,11]. As RNA:DNA hybrids and miRNAs can act to modulate gene expression, we chose to examine how a loss of XRN2 can influence the global transcriptional profile of gliomas. We exposed LN229 and U251 GBM cells to control and XRN2 siRNAs. Transfection of XRN2 siRNA resulted in a ~80% decrease in XRN2 expression in both LN229 and U251 cells lines (Figure 5A). Using RNA sequencing, we found substantial gene expression loss and gains after XRN2 loss (Figure 5B), with 194 of these genes demonstrating similar expression changes in both U251 and LN229 cell lines (Figure 5C). Using ingenuity pathway analysis, we found five biological pathways (cell cycle, cellular assembly and organization, cellular movement, DNA replication, recombination, and repair and cellular development) most affected by XRN2 loss (Figure 5D). These pathways are commonly used by tumor cells in the process of metastasis. 

Interestingly, one of the genes identified as having a positive correlation with XRN2 was vinculin (VCL). VCL has been shown to promote tumor progression in GBMs and prostate cancer [30,31]. Thus, we examined how a loss of VCL affected the speed and displacement of GBMs. We treated U251 cells with control, VCL, or XRN2 siRNA, along with CytoD. We found that cells treated with either VCL or XRN2 siRNAs displayed a ~30% decrease in speed (Figure 6A) and a ~40% decrease in displacement (Figure 6B), as compared to cells exposed to control siRNA. There was no significant difference in the speed or displacement in cells transfected with XRN2 or VCL siRNA. Additionally, the decrease in speed or displacement in VCL or XRN2 siRNA was not the same as the decrease seen in cells treated with CtyoD (Figure 6A,B). Interestingly, these observations were made even though XRN2 siRNAs led to a ~50% reduction in VCL expression, while VCL directed siRNAs resulted in a ~90% reduction (Appendix A). These data suggest that XRN2 plays a role in cell motility and invasion in order to mediate the expression of genes in these processes. 

## 4. Discussion

One of the major obstacles in treating GBMs is the dissemination of tumor cells from the primary site to the secondary site within the brain [4]. In addition, secondary site formation of recurrent glioblastomas often occurs in areas of the brain which are inaccessible to surgical resection [5]. As these recurrent GBMs also acquire resistance to radio- and chemo-therapies, a second major obstacle [5] is that treatment options are limited for these patients. Thus, understanding the mechanisms that mediate the dissemination of glioblastoma tumor cells from the primary site to secondary sites can help improve overall patient survival. 

### 4.1. Role of XRN2 in Glioblastoma Progression

Dissemination of tumors cells from a primary site to a secondary site is a multiple-step process that depends on several cellular characteristics, including (1) the cell’s inherent ability to move away from the primary tumor site (cell motility) and (2) the cell’s ability to migrate through a solid matrix and confined spaces (invasion) [32].

In our current study, we found that a loss of XRN2 reduces cell motility by ~25% in glioblastoma cells when compared to control cells. While not essential for cell motility, XRN2 is required for efficient glioblastoma cellular movement. In addition, we found that XRN2 loss also reduces migration through a matrix by ~50% as compared to the control cells. These results are consistent with observations made in lung cancer, i.e., that XRN2 is required for tumor cell invasion [13].

As we have found that XRN2 mediates dissemination of glioblastomas, XRN2′s mechanistic role in this process remains unclear. In lung cancer, it has been suggested that XRN2 regulation of miR-10a is responsible for the migration process [13]. We found that a loss of XRN2 alters the transcriptional profile of U251 and LN229 glioblastoma cells. Yet, we did not find changes in miR-10a in glioblastoma cells. One possibility for this may be that the next-generation sequencing we performed was not sensitive enough to distinguish changes in miRNAs.

One intriguing reason for the change in the glioblastoma transcriptome is the formation of RNA:DNA hybrids or R-loops. R-loops can mediate the expression of genes through a variety of different mechanisms, such as regulating the methylation status of CpG island in the promoter region of genes [33]. We have found that a loss of XRN2 results in increased R-loop formation in glioblastoma cells [10]. A loss of XRN2 has also been shown to lead to changes in the expression of genes through an R-loop-dependent manner [12]. These data suggest that R-loop regulation plays a role in mediating cell motility and invasion. 

### 4.2. XRN2 as a Potential Target for Glioblastoma Therapy

We provide evidence that XRN2 is important for glioblastoma maintenance in vivo. XRN2 expression is increased in glioblastoma patient samples as compared to normal brain. This increase in XRN2 expression also correlates with poor overall patient survival. In addition, XRN- deficient G55 cells also fail to form tumors when injected into mouse brains. Our current study demonstrates that XRN2 is required for the efficient dissemination of glioblastoma cells. Interestingly, we have previously shown that XRN2 also plays a role in the DNA damage response [10,11]. A loss of XRN2 impairs the two major double-strand break repair pathways: non-homologous end-joining and homologous recombination [10,11]. XRN2 deficiency also results in increased sensitivity to several different types of genotoxic stress, such as ionizing radiation and PARP1 inhibitors [10,11]. Our data suggest that measuring XRN2 levels in GBMs can help to determine how a patient will respond to radiation or chemotherapy. This is especially important considering that we found the highest XRN2 levels in GBMs, even when compared to other brain malignancies. Thus, XRN2 may be an important biomarker in the treatment of patients suffering from GBMs. 

As previously mentioned, there are two major obstacles in treating GBM patients: (1) the recurrence of GBMs at a secondary/un-operatable area of the brain and (2) the acquisition of therapeutic resistance in the recurrent GBMs. Targeting XRN2 can impair both of the major obstacles. First, our current study demonstrates that targeting XRN2 can impair GBM cell motility and invasion, which can then limit the potential for secondary site formation. Second, targeting XRN2 can also enhance anti-GBM therapies, especially radiation, which is a first-line anti-GBM therapy [5], as well as PARP1 inhibitors, which are currently used in clinical trials for GBM therapy [34].

## Figures and Tables

**Figure 1 cells-11-01481-f001:**
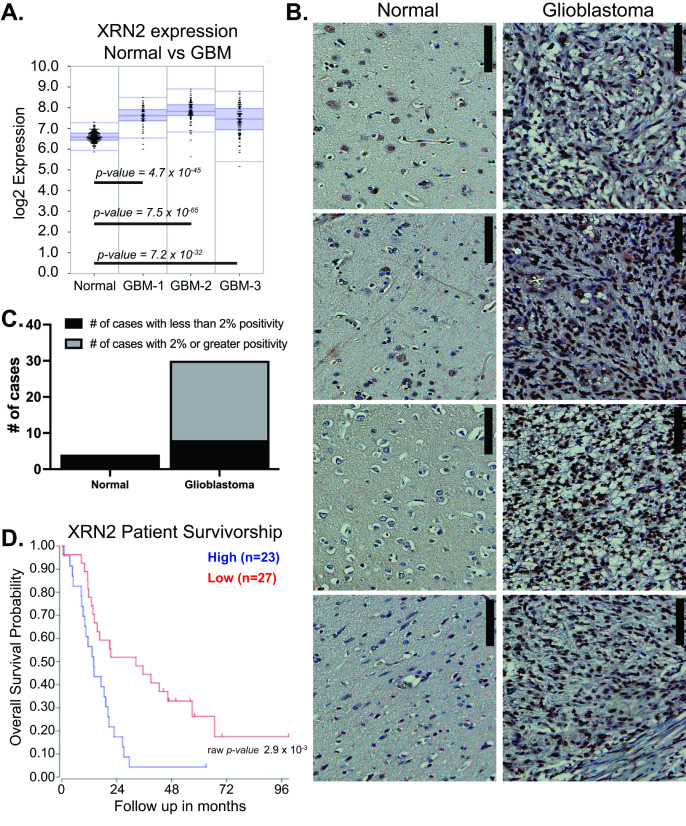
XRN2 expression confers with glioblastoma disease and poor patient survival. (**A**) Log2 XRN2 mRNA expression levels in normal (Berchtold, 172 samples), GBM-1 (Pfister, 46 samples), GBM-2 (Loeffler, 70 samples), and GBM-3 (Hegi, 84 samples) databases. Graph generated from R2: Genomics Analysis and Visualization Platform. (**B**) Representative XRN2 immunohistochemistry staining of normal and glioblastoma patient samples. Brown signal is XRN2. Blue signal is hematoxylin (nuclei). Scale bar is 100 micron. (**C**) Quantification of XRN2 signal from immunohistochemistry staining of normal and GBM patient samples. A signal of less than 2% is considered negative, while a signal of 2% or greater is considered positive. (**D**) Glioma patient survival outlook based on XRN2 mRNA levels (Kawaguchi, 50 patients). Blue line denotes patients with high expression of XRN2. Red line denotes patients with low expression of XRN2. Graph generated from R2: Genomics Analysis and Visualization Platform. Statistical analysis was performed using the default setting provided by platform.

**Figure 2 cells-11-01481-f002:**
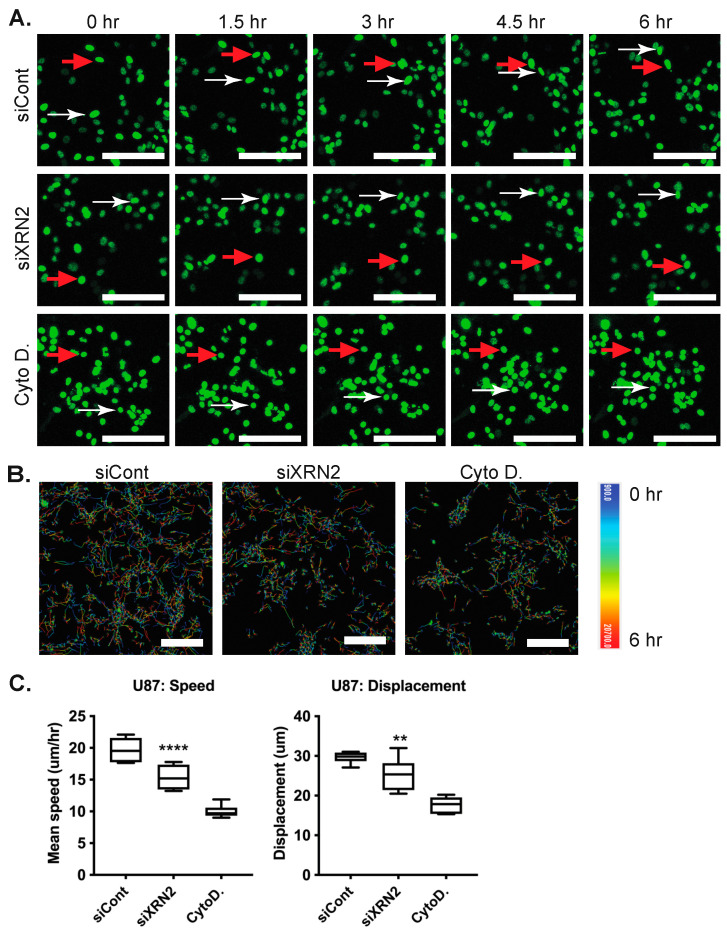
XRN2 is required for GBM cell motility. (**A**) Image stills from live-cell imaging of U87-GFP cells. Arrows track the movement of two representative cells over time. Tracking was over 6 h with images taken at 30-min intervals. (**B**) Positional tracks of U87 H2B-GFP cells during the 6 h live-cell imaging. White marks the position of the cells at the beginning of time to red, the position of the cells at the end of imaging. (**C**) Quantification of U87-GFP tracking. Changes in speed and displacement upon XRN2 knockdown by siRNA are shown. Scale bar is 100 micron. ** *p*-value ≤ 0.01, **** *p*-value ≤ 0.0001. The Students *t*-test was used for statistical analysis.

**Figure 3 cells-11-01481-f003:**
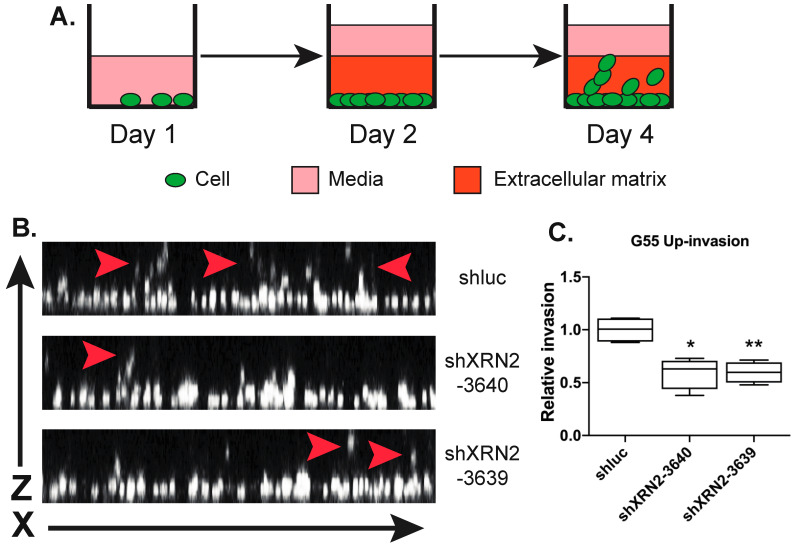
XRN2 is required for invasion through a matrix. (**A**) Diagram of inverted vertical invasion. Day 1—cells are plated at near confluent levels. Day 2—extracellular matrix is applied to the confluent cells. Day 4—cultures are fixed, stained, and imaged for invasion analysis. (**B**) Representative ZX image of invading G55 cells (white). Red arrows denote invading cells. (**C**) Quantification of up-invasion of G55 cells with the listed shRNAs. * = *p*-value ≤ 0.05, ** = *p*-value ≤ 0.01. The Students *t*-test was used for statistical analysis.

**Figure 4 cells-11-01481-f004:**
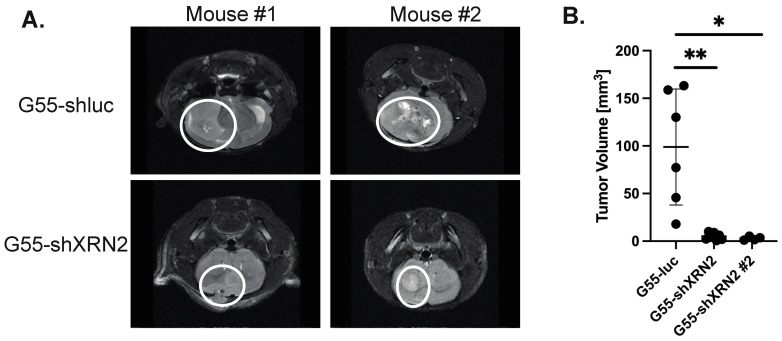
Loss of XRN2 results in decreased tumor volume in vivo. (**A**) Magnetic resonance imaging was used to detect control (G55-shluc) and XRN2-deficient (G55-shXRN2) tumors in mouse brains. (**B**) Quantitation of tumor volumes obtained from control (G55-shluc) and two unique XRN2-deficient (G55-shXRN2 and G55-shXRN2 #2) G55 cell lines. * = *p*-value ≤ 0.05, ** = *p*-value ≤ 0.01. The Students *t*-test was used for statistical analysis.

**Figure 5 cells-11-01481-f005:**
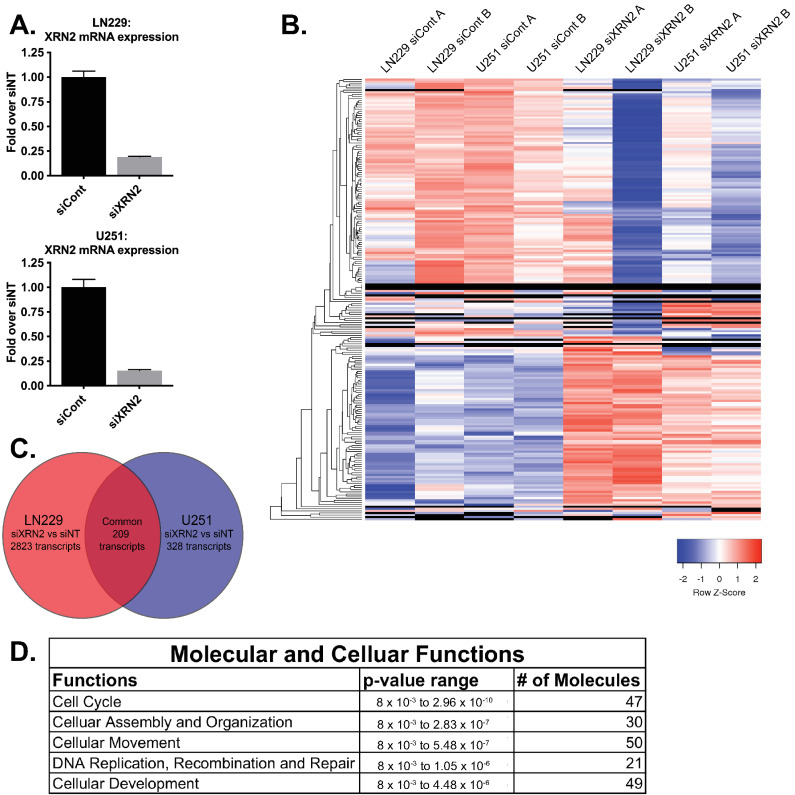
XRN2-mediated transcriptome landscape. (**A**) qPCR of XRN2 expression of samples used in the RNA-Seq in B. (**B**) RNA-Seq heat map of LN229 and U251 transfected with siCont or siXRN2. Heat map generated from transcripts with a log 1.3 or greater change and a *p*-value of 0.05 or better. (**C**) Venn diagram of overlapping transcripts from B. (**D**) Ingenuity pathway analysis of the overlapping transcripts in C.

**Figure 6 cells-11-01481-f006:**
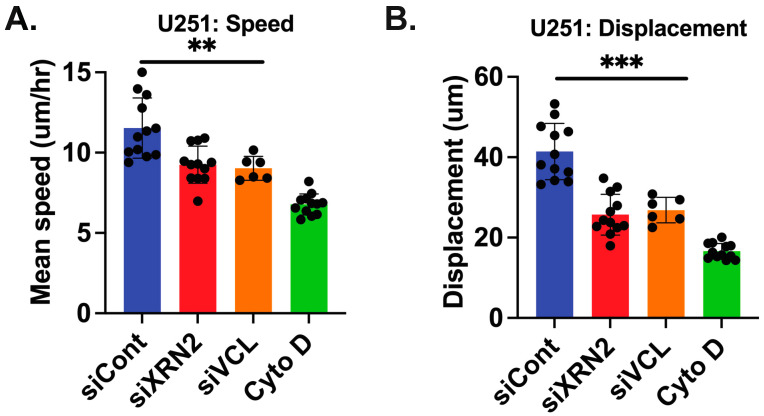
Loss of VCL or XRN2 results in a similar decrease in the speed or displacement in GBM cells. Quantification of U251-GFP tracking. Changes in (**A**) speed and (**B**) displacement upon control, VCL or XRN2 knockdown by siRNA are shown. ** *p*-value ≤ 0.01, *** *p*-value ≤ 0.001. The Students *t*-test was used for statistical analysis.

## Data Availability

Links to public data sources and RNA-seq data are listed in methods.

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
