# Peer review of "XRN2 Is Required for Cell Motility and Invasion in Glioblastomas"

_cells, 2022, doi:10.3390/cells11091481_

Round 1

Reviewer 1 Report

The manuscript by Dang TT et al. describes the finding that XRN2 gene product expression is important for cell motility and invasion of glioblastoma (GBM) cells. XRN2 is a 5’- 3’ exoribonuclease which affects many different cellular processes such as maturation of ribosomal RNA, transcription termination, RNA polymerase pausing and premature termination as well as DNA repair. Previous studies indicated that XRN2 promotes epithelial to mesenchymal transition (ETM) in lung cancer cells and suggested that XRN2-dependent enhancement in maturation of miR-10a was responsible for ETM inhibition in lung cancer.

To investigate the role of XRN2 in GBMs Dang et al. start from the analysis of glioma gene expression data set and find that XRN2 mRNA expression is higher in GBM with respect to normal brain tissue and that higher XNR2 correlates with reduced patient overall survival. They also demonstrate that 1) XRN2 protein expression correlates with the grade of glioma 2) XRN2 downregulation correlates with in vitro reduced cell motility and invasion and 3) with in vivo tumor growth in orthotopic xenografts.

To investigate possible molecular mechanisms, Dang et al. quantified, by RNA-seq, gene expression profiles of GBM cells in the presence or absence of XRN2. By bioinformatic analysis of the genes differentially expressed upon XRN2 loss, they identify five biological pathways (cell cycle, cellular assembly and organization, cellular movement, DNA replication recombination and repair, cellular development) are those predicted to be the most affected. Among the differentially regulated genes they focus on vinculin whose expression positively correlates with XRN2 and demonstrate that downregulation vinculin or XRN2 expression via RNA interference, reduces speed and displacement of GBM cells to a similar extent.

Although XRN2 role in cancer has been described before, this is the first report of its upregulation in GBM, and as such deserves publication in “Cells”. There are however a few points which should be clarified.

  • The experimental work is based on a different established human GBM cell lines which are employed in different experimental settings: U87 and U251 are used to investigate motility in vitro, G55 and XRN2-deficient G55 to measure cellular invasion in vitro, G55 for grafting in vivo, LN299 and U251 for comparing transcriptional profiles in XRN2 upon XRN2 silencing and U251 again for comparing the effect of XRN2 or vinculin inhibition. However, GMB are quite heterogenous tumor and established GBM cell lines may substantially differ one from the other. Can the author comment on this point?  
  • The experiment with orthotopically grafted cells shows that loss of XRN2 diminishes GBM growth in vivo. This finding suggests that XRN2 is important for tumor cell division or cell death inhibition but does not bring any support to its function in cell migration or invasion. A role of XRN2 in cell cycle is also compatible with the RNA-seq analysis (Figure 5D). Is G55 cell growth inhibited also in vitro? If so, the authors should show it, if not, they should discuss this point.
  • In Figure 4 tumor images from two mice are shown. It should be mentioned how many animals were used in total.
  • Which new information on XRN2 role in motility and invasion do the findings described in Figure 6 provide? Which is the extent of inhibition in vinculin expression caused by loss of XRN2? Is it comparable to the one achieved by interfering oligos?

Minor points

The sequences of oligos used in vinculin interference should be provided in the method section as well as the extent of vinculin inhibition.

Lane 185 reference 16 and 28 do not seem appropriate (do not relate to lung cancer or oral carcinoma)

Lane 231 ……. U251 GBM cells (Supplemental Figure 2 C) (in the Figure legend they are called LL229)

Author Response

We would like to thank this reviewer for their comments on our manuscript. Please find a point-by-point response (yellow text) to comments:

The experimental work is based on a different established human GBM cell lines which are employed in different experimental settings: U87 and U251 are used to investigate motility in vitro, G55 and XRN2-deficient G55 to measure cellular invasion in vitro, G55 for grafting in vivo, LN299 and U251 for comparing transcriptional profiles in XRN2 upon XRN2 silencing and U251 again for comparing the effect of XRN2 or vinculin inhibition. However, GMB are quite heterogenous tumor and established GBM cell lines may substantially differ one from the other. Can the author comment on this point?  

Response: Thank you for the comment. The heterogenous nature of GBMs is the exact reason why we choose to use multiple different cell lines which are also heterogenous to each other. Finding consistent data among the cell lines suggest that these results may be more applicable to patient samples rather than observing them in a single cell line.

The experiment with orthotopically grafted cells shows that loss of XRN2 diminishes GBM growth in vivo. This finding suggests that XRN2 is important for tumor cell division or cell death inhibition but does not bring any support to its function in cell migration or invasion. A role of XRN2 in cell cycle is also compatible with the RNA-seq analysis (Figure 5D). Is G55 cell growth inhibited also in vitro? If so, the authors should show it, if not, they should discuss this point.

Response: We would like to thank the reviewer for their comment. We have previously published that the loss of XRN2 does not alter the proliferation rate or cell cycle profile of these cells in tissue culture setting. Also, others have also published that loss of XRN2 does not adversely affect proliferation rates of cells. Appropriate citations have been included. Lines 282-285

In Figure 4 tumor images from two mice are shown. It should be mentioned how many animals were used in total.

Response: Yes, thank you for this comment. The number of mice injected for each condition has been added to text. Line 275-276

Which new information on XRN2 role in motility and invasion do the findings described in Figure 6 provide? Which is the extent of inhibition in vinculin expression caused by loss of XRN2? Is it comparable to the one achieved by interfering oligos?

Response: This is an important question. We have found that XRN siRNA led to a greater than 50% reduction in vinculin expression. Use of vinculin siRNA led to ~90% reduction. This data has been included as supplemental data 6. Lines 322-324

Minor points

The sequences of oligos used in vinculin interference should be provided in the method section as well as the extent of vinculin inhibition.

Response: Sequence of oligos has been added to methods section and extent of loss has been added to supplemental data.

Lane 185 reference 16 and 28 do not seem appropriate (do not relate to lung cancer or oral carcinoma)

Response to reviewers: We would like to thank the reviewer for pointing out this oversight. Citations have been corrected.

Lane 231 ……. U251 GBM cells (Supplemental Figure 2 C) (in the Figure legend they are called LL229)

Response to reviewers: The figure legend has been corrected and now reads U251.

Reviewer 2 Report

Glioblastoma multiforme (GBM) are highly invasive tumors that spread through the brain causing secondary tumors. High invasiveness coupled with the acquisition of therapy resistance makes these tumors very hard to treat. Thus, the understanding molecular mechanism driving invasiveness in GBM is of immense importance. In the manuscript titled “XRN2 is required for cell motility and invasion in glioblastomas”, the authors Dang et al. study the role 5’-3’ exoribonuclease XRN2 plays in driving invasive phenotype in GBM. Based on an analysis of multiple published datasets, the authors demonstrate that XRN2 expressed at a higher amount in GBM. Using cell biological assays, the authors demonstrate that XRN2 is crucial for increased cellular speed, displacement, and invasion through the extracellular matrix. Additionally, the authors demonstrate that XRN2 expression in the GBM cell line is important for xenograft tumor formation. Finally, using RNA sequencing, the authors show that loss XRN2 leads to alteration in expression of genes related to cell cycle, cellular assembly, cellular movement, etc.  The role of XRN2 in invasive phenotype GBM cells is novel to the best of my knowledge. The manuscript is well written and attempts to gain insight into an important aspect of GBM biology. The manuscript will be suitable for the readership of Cells. However, in its present form, the manuscript has several methodological issues listed below that need to be addressed before publication in Cells.  1. The authors use different GBM cell lines for different experiments throughout the manuscript. For example, U87 cells in fig.2, G55 cells in fig.3 and 4, LN229 and U251 for fig.5, and U251 for fig.6. Do all GBM cell lines show the same magnitude of effect? Why different cell lines are chosen for different experiments? 2. The authors use siRNA or shRNA to knock down the expression of XRN2 for most of the experiments. Both siRNA and shRNA are susceptible to off-target effects. The authors must include rescue experiments to show that the observed effect is indeed due to XRN2 depletion. 3. It is unclear which statistical test was used for data analysis throughout the paper. This information should be included in the manuscript. 4. In fig. 3C, the authors should clarify what they mean by "relative invasion", and how was it calculated. 5. In fig. 4, the authors show that loss of XRN2 leads to reduced tumor volume in vivo. This phenotype can be independent of invasiveness and may be driven by reduced cell growth or increased cell death. The authors should determine if the doubling time of G55 cells is affected upon XRN2 depletion? 6. In supplemental fig. 4, the authors show the images of xenograft tumors formed in the mouse brain. The tumors formed G55 cells appear to be a single mass in both the mouse. In contrast, the tumor formed by the G55 cell expressing shXRN2 shows two completely separated masses in mouse 1 and two large masses connected by a thin layer of cell in mouse 2. This observation seems contrary to the claims made by the authors about the role of XRN2 in increasing the invasiveness of the GBM cells. The authors should clarify this discrepancy. 7. In fig. 5, the authors use RNA sequencing of two GBM cell lines to uncover the XRN2 mediated transcription landscape. Panel C shows that the cell line to cell line transcription variation is immense amongst different GBM cell lines. This is compounded by the fact that the authors did not use any of these cells for their earlier phenotypic analysis. This makes it almost impossible to correlate the RNAseq results with that of the phenotypic characterization. To make the study comprehensible the authors should repeat the phenotypic characterization (cell motility and invasion) with LN229 and U251 cell lines.

Author Response

We would like to thank this reviewer for their comments, please find a point-by-point response to comment (Yellow text):

The authors use different GBM cell lines for different experiments throughout the manuscript. For example, U87 cells in fig.2, G55 cells in fig.3 and 4, LN229 and U251 for fig.5, and U251 for fig.6. Do all GBM cell lines show the same magnitude of effect? Why different cell lines are chosen for different experiments?

Response: Different cell lines were chosen for different experiments only to demonstrate that loss of XRN2 can affect multiple different cells lines as compared to single line itself. The only cell line specifically chosen for a particular experiment was G55 for the xenograft experiments. Dr. Towner, who was performing the MRI experiments recommended these cells as they are the ones he had most experience with and MRI is quite costly.

The authors use siRNA or shRNA to knock down the expression of XRN2 for most of the experiments. Both siRNA and shRNA are susceptible to off-target effects. The authors must include rescue experiments to show that the observed effect is indeed due to XRN2 depletion.

Response: We understand the reviewers concerns. We attempted to re-express XRN2 in XRN2 depleted cells, but could never achieve wild type levels. We do feel that this concern is mitigated by the fact that we used both siRNA and shRNA technologies, with different targeting sequences, in parallel with one another in multiple different cell lines and observed consistent results. 

It is unclear which statistical test was used for data analysis throughout the paper. This information should be included in the manuscript.

Response: Thank for the comment, we have added tests used to figure legends.

In fig. 3C, the authors should clarify what they mean by "relative invasion", and how was it calculated.

Response: How relative invasion was calculated has been added to manuscript. Lines 259-261

In fig. 4, the authors show that loss of XRN2 leads to reduced tumor volume in vivo. This phenotype can be independent of invasiveness and may be driven by reduced cell growth or increased cell death. The authors should determine if the doubling time of G55 cells is affected upon XRN2 depletion?

Response: We would like to thank the reviewer for their comment. We have previously published that the loss of XRN2 does not alter the proliferation rate or cell cycle profile of these cells in tissue culture setting. Also, others have also published that loss of XRN2 does not adversely affect proliferation rates of cells. Appropriate citations have been included. Lines 282-285

In supplemental fig. 4, the authors show the images of xenograft tumors formed in the mouse brain. The tumors formed G55 cells appear to be a single mass in both the mouse. In contrast, the tumor formed by the G55 cell expressing shXRN2 shows two completely separated masses in mouse 1 and two large masses connected by a thin layer of cell in mouse 2. This observation seems contrary to the claims made by the authors about the role of XRN2 in increasing the invasiveness of the GBM cells. The authors should clarify this discrepancy.

Response: We would like to thank the reviewer for this astute comment. We did notice this as well, but we would like to point that these injections were performed using pooled XRN2 knockdown cell lines. Thus, we contend that what is seen is this image are the limited amount of cells in this pool that retained enough XRN2 expression to have movement.

In fig. 5, the authors use RNA sequencing of two GBM cell lines to uncover the XRN2 mediated transcription landscape. Panel C shows that the cell line to cell line transcription variation is immense amongst different GBM cell lines. This is compounded by the fact that the authors did not use any of these cells for their earlier phenotypic analysis. This makes it almost impossible to correlate the RNAseq results with that of the phenotypic characterization. To make the study comprehensible the authors should repeat the phenotypic characterization (cell motility and invasion) with LN229 and U251 cell lines.

Response: We would like to thank the reviewer for their comment. Data demonstrating loss of cell motility after XRN2 loss is provided in supplemental Figure 2. Data demonstrating loss of invasion in LN229 cells has been provided in supplemental data.

Round 2

Reviewer 1 Report

In the revised version the authors respond to most of the risen issues. However, they do not provide an explanation for the differences between the effects of XRN2 in vitro (promoting cell invasion) and in vivo (reducing cell growth). As a consequence, it is hard to have an insight in mechanism through which XRN2 contribute to GBM aggressivity and the work remains essentially descriptive. Nevertheless, since the finding is interesting the paper is worth of publication.

Reviewer 2 Report

The authors have addressed my concerns for the most part in the revised manuscript.